# Changing Laboratory Practice for Early Detection of a Fetal Inflammatory Response: A Contemporary Approach

**Yin Ping Wong** [1,2,*] and **T Yee Khong** [2,3]

1 Department of Pathology, Faculty of Medicine, Universiti Kebangsaan Malaysia, Kuala Lumpur 56000, Malaysia

2 Department of Pathology, SA Pathology, Women's and Children's Hospital, North Adelaide, SA 5006, Australia

3 Department of Pathology, Faculty of Health and Medical Sciences, University of Adelaide, Adelaide, SA 5000, Australia

* Correspondence: ypwong@ppukm.ukm.edu.my; Tel.: +60-3-91455364

**Abstract:** Neonates born with the fetal inflammatory response (FIR) are at risk of complications such as early-onset neonatal sepsis, meningitis, and pneumonia. Providing an early histopathological diagnosis of FIR is important to guide management but can be a challenge in busy laboratories. This is a retrospective cross-sectional study over a four-month duration recruiting all placental cases with histological chorioamnionitis in our institution. The diagnostic performance of the umbilical cord (UC) section in identifying FIR, relative to the corresponding subsequent placental sections, was assessed. Clinical predictors of umbilical cord FIR were also investigated. A total of 390 UC sections were analyzed, of which 206 (52.8%) were found positive for FIR: 111 cases (53.9%) stage 1, 87 (42.2%) stage 2, and 8 (3.9%) stage 3. Our data revealed a good diagnostic sensitivity, specificity, positive predictive value, and accuracy of 76.2% (95%CI: 68.6–82.7%), 82.4% (95%CI: 65.5–93.2%), 95.0% (95%CI: 90.2–97.6%), and 77.3% (95%CI: 70.6–83.1%) respectively, in cases when clinical chorioamnionitis, fever and/or prolonged rupture of membrane (PROM) were suspected, with the area under the curve of 0.793. A maternal inflammatory response (MIR) was correlated with FIR ($p < 0.001$). Multivariate logistic regression analysis indicated that the higher the gestational age, clinical suspicion of chorioamnionitis, fever, and/or PROM, and the higher the stage of MIR significantly increased the odds of FIR ($p < 0.001$). UC section diagnosis of FIR is reasonably accurate in cases with clinical chorioamnionitis, fever, and/or PROM. Changing current laboratory practice to rapid processing of UC ahead of the rest of the other placental sections can be recommended in busy pathology departments.

**Keywords:** chorioamnionitis; COVID-19; fetal vasculitis; funisitis; intra-amniotic infection; placenta; umbilical cord

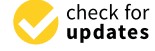



## 1. Introduction

Anatomical pathology laboratories are under increasing pressure to provide timely and accurate results while the workload and complexity of specimens and diagnostic methodologies have increased. Thus, laboratories must continually modify or adapt their protocols. Confirmation of fetal inflammatory response (FIR) to an intra-amniotic infection is important as FIR is implicated in serious neonatal morbidities such as early and late-onset neonatal sepsis, meningitis, and pneumonia. Thus, neonates born with a clinical suspicion of intra-amniotic infection are started on empirical antibiotic therapy in many cases while awaiting laboratory confirmation [1]. Conventionally, the umbilical cord, extraplacental membranes, and placental discs are processed together, and this routine processing can take a while because of the need for adequate fixation of the highly vascular placental tissue. Previous methods for rapid diagnosis of intra-amniotic infection, such as placental smears of the chorioamniotic plate and the fetal side of the extraplacental membranes [2,3]

or frozen sections of the umbilical cord [4], have a relatively low diagnostic sensitivity and specificity, with high false negative and positive rates and are disruptive to the routine workflow of an anatomical pathology laboratory. We report a validation of a contemporary approach to early confirmation of the presence or absence of FIR by an expedited processing of the umbilical cord (UC) before the rest of the placenta.

## 2. Materials and Methods

The study was approved by the Human Research Ethical Committee of the Institutional Review Board (1287A/09/2025). All cases with a histological diagnosis of chorioamnionitis from pregnancies delivered in Women's and Children's Hospital during the period of March 2022 to June 2022 were included in the study. Indications on the histopathology requisition form were recorded. Histological slides were reviewed for the fetal inflammatory response in the umbilical cord (UC) section and in the placental sections separately, and for maternal inflammatory response, blinded to the original clinical information and histological diagnosis, according to the 2016 Amsterdam Placental Workshop Group Consensus guidelines [5].

The value of the UC section in identifying FIR was evaluated relative to the final histopathological diagnosis following examination of all sections of the placenta (considered as the gold standard in this study). Sensitivity, specificity, positive predictive value (PPV), and negative predictive value (NPV) for UC diagnosis of FIR, relative to corresponding placental sections, were calculated. Cohen's Kappa coefficient statistic was used to measure the reliability of the UC section in detecting FIR. Diagnostic efficacy and validity of the UC section were determined by receiver operating characteristic (ROC) curve analysis. Chi-square, Fisher exact test, and Mann–Whitney U test were performed to compare the differences between variables. Logistic regression analysis was conducted to evaluate the relationship between various predictor variables and FIR, controlling for the effects of other potential confounders. Statistical analyses were carried out using the Statistical Package for the Social Science (SPSS) software version 26.0 (PASW Statistics, USA). A *p*-value of less than 0.05 was considered statistically significant.

## 3. Results

Three hundred and ninety (390) of the 1680 placentas received with various indications for histopathological examination had a histological diagnosis of chorioamnionitis with 185 (47.4%) of them recording clinical suspicion of chorioamnionitis, intrapartum maternal pyrexia and/or prolonged rupture of membrane (PROM) as the indications.

FIR was found in 206 (UC FIR+) of the 390 UC from placentas with histological chorioamnionitis: 111 (53.9%) were stage 1, 87 (42.2%) were stage 2 and 8 (3.9%) were stage 3. Indications of clinical suspicion of chorioamnionitis, intrapartum maternal pyrexia and/or prolonged rupture of membrane (PROM) were recorded in 121 of the 206 UC FIR+ cases. Other indications in the remaining 85 UC FIR+ cases were maternal COVID-19 (n = 38), fetal distress (FD)/meconium-stained liquor (MSL) (n = 9), maternal vascular malperfusion-type indications including gestational hypertension (n = 6), fetal growth restriction (n = 8) and abruption (n = 7), gestational diabetes mellitus (n = 3), retained placenta (n = 10) and other maternal indications such as no antenatal care and poor obstetric history (n = 4). Clinicopathological characteristics of the study population in relation to UC inflammation are summarised in Table 1.

**Table 1.** Clinicopathological characteristics of the study population.

| Clinicopathological Parameters | Cord Inflammation n = 206 (%) | No Cord Inflammation n = 184 (%) | *p* Value |
|---|---|---|---|
| Gestational age, median weeks (range) | 39.0 (16.6–41.9) | 38.7 (17.0–41.4) | 0.013 * |
| CA/fever/PROM | | | <0.001 * |
| Yes | 121 (58.7) | 64 (34.8) | |
| No | 85 (41.3) | 120 (65.2) | |
| Maternal inflammatory response | | | <0.001 * |
| Stage 1 | 49 (23.8) | 121 (65.8) | |
| Stage 2 | 144 (69.9) | 61 (33.1) | |
| Stage 3 | 13 (6.3) | 2 (1.1) | |
| ¥ Fetal inflammatory response | | | <0.001 * |
| Stage 0 | 18 (8.7) | 87 (47.3) | |
| Stage 1 | 188 (91.3) | 97 (52.7) | |

Abbreviations: CA = clinical chorioamnionitis; PROM = prolonged rupture of membrane; * statistically significant; ¥ corresponding sections of the placenta.

A comparison of the FIR status in the UC and the corresponding placental sections revealed that 188 cases with FIR and 87 cases without FIR in the UC were concordant with those of the corresponding placental sections, with a concordant rate of 70.5%. In 97 cases, FIR was not seen in the UC but was detected in the corresponding placental sections. Clinical indications for placental examination for these 97 cases were clinical suspicion of chorioamnionitis, fever, and/or PROM in 36 and for other indications in the remaining 61 cases (Figure 1). Among these 22 cases of COVID-19, the majority were term pregnancies and in stage 1 MIR; only one delivered preterm at 35-week gestation and in stage 2 MIR.

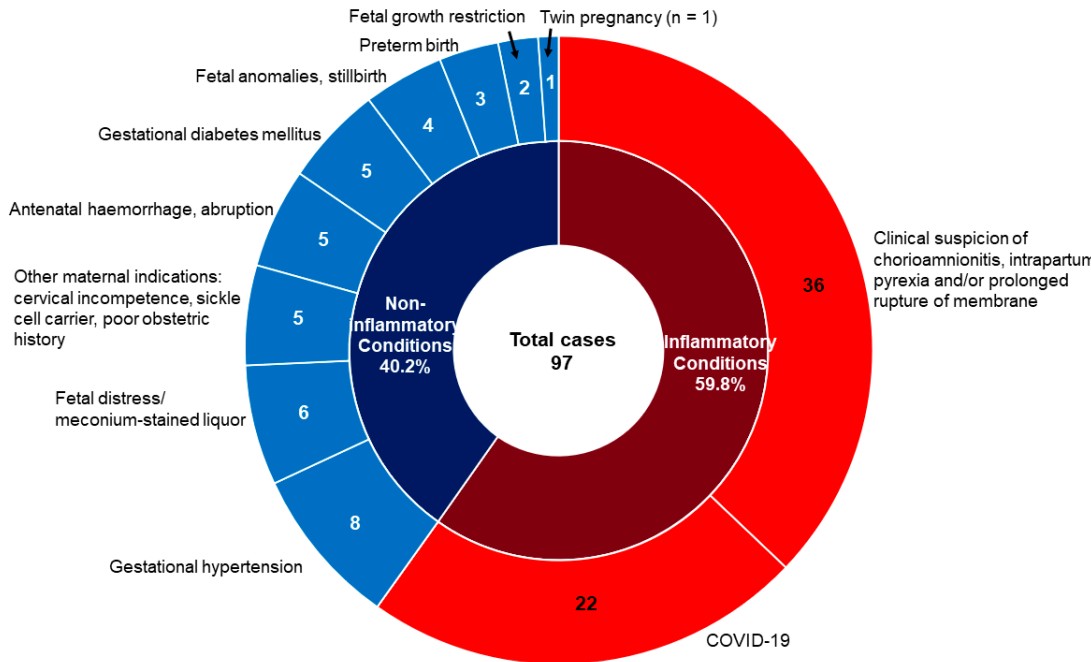

**Figure 1.** Clinical indications for placental examination in 97 cases with FIR detected in corresponding placental sections but not in the umbilical cord section. Inner circle: cases classified as inflammatory (n = 58) or non-inflammatory (n = 39) conditions; Outer circle: red arc area refers to the 58 inflammatory cases and the blue arc area refer to the 39 non-inflammatory conditions.

There was a significant association between the presence of umbilical cord FIR and the positive MIR status in the corresponding placental sections ($p < 0.001$) (Table 1). Almost half (n = 184, 47.2%) of the cases revealed MIR, of which 63 (34.2%) were Stage 2 or 3, despite the absence of umbilical cord FIR. Multivariate analysis indicated that the higher the gestational age, clinical suspicion of chorioamnionitis, fever, and/or PROM, and the higher the stage of MIR significantly increased the odds of FIR (Table 2).

**Table 2.** Clinical predictors of the fetal inflammatory response by multivariate logistic regression.

| Clinical Parameters | Odd Ratios | 95%CI | *p* Value |
|---|---|---|---|
| Gestational age, weeks | 1.084 | 1.028–1.143 | 0.003 * |
| CA/fever/PROM | 2.475 | 1.560–3.928 | <0.001 * |
| Maternal inflammatory response | | | |
| MIR1 (reference) | | | <0.001 * |
| MIR2 | 5.764 | 3.610–9.204 | <0.001 * |
| MIR3 | 21.146 | 4.307–103.823 | <0.001 * |

Abbreviations: CA = clinical chorioamnionitis; CI = confidence interval; MIR = maternal inflammatory response; PROM = prolonged rupture of membrane; * statistically significant.

To refine our analysis, the 390 placentas with histological chorioamnionitis were further divided into two groups according to the clinical indications: group 1, placentas being sent for clinical suspicion of chorioamnionitis, fever and/or PROM (n = 185); group 1a, cases from group 1 plus those being sent for COVID-19 or fetal distress/meconium-stained liquor (MSL) (n = 284); and group 2, placentas being sent for indications other than those of group 1 (n = 205). Table 3 compares UC FIR results with the status of inflammation (FIR and MIR) of the corresponding placental sections among the three groups.

**Table 3.** Comparison of placental inflammation between the three groups with distinct clinical indications.

| Corresponding Placental Sections | | Umbilical Cord FIR | | | | | |
|---|---|---|---|---|---|---|---|
| | | Group 1 (n = 185) | | Group 1a (n = 284) | | Group 2 (n = 205) | |
| | | FIR Stage 1, 2 or 3 n = 121 (%) | No FIR n = 64 (%) | FIR Stage 1, 2 or 3 n = 166 (%) | No FIR n = 118 (%) | FIR Stage 1, 2 or 3 n = 85 (%) | No FIR n = 120 (%) |
| FIR | No FIR | 6 (5.0) | 28 (43.8) | 12 (7.2) | 54 (45.8) | 12 (14.1) | 59 (49.2) |
| | Stage 1 | 115 (95.0) | 36 (56.3) | 154 (92.8) | 64 (54.2) | 73 (85.9) | 61 (50.8) |
| MIR | Stage 1 | 25 (20.7) | 39 (60.9) | 39 (23.5) | 77 (65.3) | 24 (28.2) | 82 (68.3) |
| | Stage 2 | 88 (72.7) | 23 (35.9) | 119 (71.7) | 39 (33.0) | 56 (65.9) | 38 (31.7) |
| | Stage 3 | 8 (6.6) | 2 (3.1) | 8 (4.8) | 2 (1.7) | 5 (5.9) | 0 (0.0) |

Abbreviations: CA = clinical chorioamnionitis; FIR = fetal inflammatory response; MIR = maternal inflammatory response; PROM = prolonged rupture of membrane; UC = umbilical cord; group 1 = cases with CA/fever/PROM; group 1a = group 1 + cases with COVID-19 and fetal distress/meconium-stained liquor; group 2 = cases with indications other than CA/fever/PROM.

The other submitted sections of placentas showed evidence of FIR in 151 (81.6%), 218 (76.8%), and 134 (65.4%) cases in group 1, group 1a, and group 2 respectively. There were moderate and fair agreements between the UC results on FIR with the other submitted sections of placentas for group 1, group 1a, and group 2, with Kappa (κ) values of 0.436, 0.412, and 0.323 ($p < 0.001$), respectively. The diagnostic performance of UC block for the detection of FIR among the three groups with distinct clinical indications is depicted in Table 4, with the ROC curves shown in Figure 2.

**Table 4.** Diagnostic efficacy of UC block in the identification of FIR in the three groups with distinct clinical indications.

| | N | Diagnostic Performance (%) | | | | |
|---|---|---|---|---|---|---|
| | | Sensitivity (95%CI) | Specificity (95%CI) | PPV (95%CI) | NPV (95%CI) | Accuracy (95%CI) |
| All Groups | 390 | 66.0 (60.1–71.5) | 82.9 (74.3–89.5) | 91.3 (87.2–94.1) | 47.3 (42.7–51.9) | 70.5 (65.7–75.0) |
| Group 1 | 185 | 76.2 (68.6–82.7) | 82.4 (65.5–93.2) | 95.0 (90.2–97.6) | 43.8 (36.0–51.8) | 77.3 (70.6–83.1) |
| Group 1a | 284 | 70.6 (64.1–76.6) | 81.8 (70.4–90.2) | 92.8 (88.4–95.6) | 45.8 (40.0–51.6) | 73.2 (67.7–78.3) |
| Group 2 | 205 | 54.5 (45.7–63.1) | 83.1 (72.3–91.0) | 85.9 (78.0–91.3) | 49.2 (43.9–54.5) | 64.4 (57.4–70.9) |

Abbreviations: CA = clinical chorioamnionitis; PROM = prolonged rupture of membrane; CI = confidence interval; PPV = positive predictive value; NPV = negative predictive value; group 1 = cases with CA/fever/PROM; group 1a = group 1 + cases with COVID-19 and fetal distress/meconium-stained liquor; group 2 = cases with indications other than CA/fever/PROM.

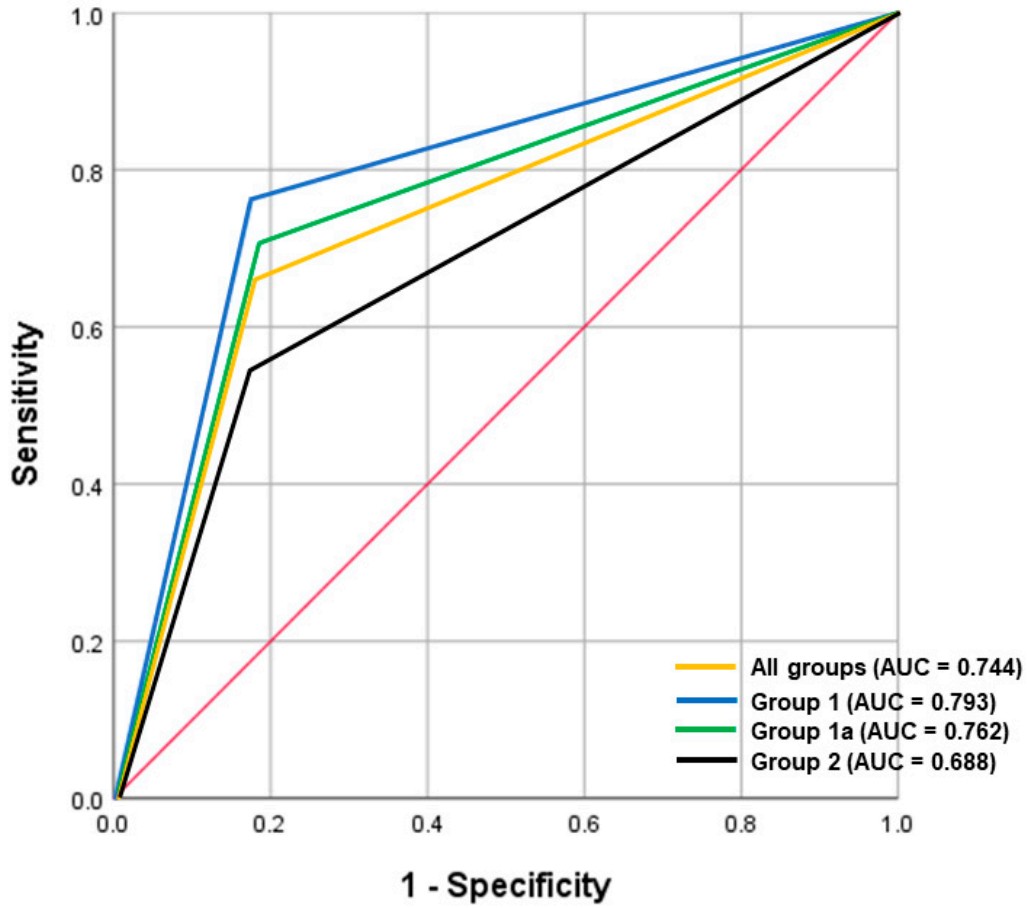

**Figure 2.** Receiver operating characteristic (ROC) analysis of the diagnostic performance of the umbilical cord section in identifying fetal inflammatory response, comparing among the three groups with distinct clinical indications.

## 4. Discussion

### 4.1. Past and Current Practice

No one test to diagnose intra-amniotic infection is perfect with 100% sensitivity and specificity. Amniotic fluid culture is the current gold standard to confirm the presence of potential intra-amniotic infectious microorganisms. Nonetheless, the use of amniotic fluid culture is limited by the long turnaround time as well as the high false negative rate owing to intrapartum antibiotics. A myriad of laboratory methods, including maternal

and cord blood plasma cytokine profile and inflammatory biomarkers such as interleukin (IL)-6, IL-8, IL-1β, tumor necrosis factor (TNF)-α, matrix metalloproteinase-8 (MMP-8), have been extensively investigated but are largely not in routine clinical use and have failed to provide an accurate ascertainment of intra-amniotic infection [6–8]. Microorganism detection by molecular methods (real-time polymerase chain reaction), although highly sensitive, could be technically challenging with potential contamination by nonpathogenic sources of DNA [9,10].

Placental smears of the chorioamniotic plate and the fetal side of the extraplacental membranes have been used for rapid diagnosis of intra-amniotic infection. With placental smears, bacterial colonies such as *E. coli*, group B streptococci and anaerobic bacteria with and/or without amniotic leukocytosis were identified. This method however had a relatively low diagnostic sensitivity and specificity, with high false negative and positive rates [2,3]. For instance, amniotic necrosis may cause potential diagnostic confusion with microorganisms.

Frozen section of the umbilical cord was also previously proposed as a valuable and rapid tool in aiding the diagnosis of intra-amniotic infection in the immediate post-partum period. To our knowledge, the frozen section technique on the umbilical cord was proposed as early as 1959 with a fair detection of cord inflammation in clinically suspected cases [4]. This method however was not widely practiced until decades later. In 2014, Mahe et al. demonstrated that the diagnostic sensitivity and specificity of the umbilical cord frozen section for FIR were 89% and 69% respectively when compared with the permanent section [11]. Although a frozen section typically could allow immediate diagnosis in less than an hour, it is disruptive to the work-flow of the laboratory and, furthermore, this technique is not without its limitations. Compared to permanent haematoxylin and eosin-stained section, the frozen section is technically more challenging in attaining good quality section and tissue staining, which will influence the interpretation by the attending pathologist. Freezing artifact for instance may cause structural distortion of tissue, jeopardizing the slide interpretation [12]. Convolution of the nuclei of the cord stromal cells may sometimes occur, closely mimicking neutrophils, leading to false positive interpretation [13].

### 4.2. The Proposed Change in the Current Practice: Evidence-Based

We conducted, to our knowledge, the first study to evaluate the diagnostic utility and validity of the UC section in the detecting FIR. This demonstrated that the UC section is reasonably accurate, with a high specificity and positive predictive value, when used as a tool for early detection of intra-amniotic infection, in a form of FIR. As expected, the diagnosis of UC FIR was most accurate with a high PPV in inflammatory conditions, namely clinical chorioamnionitis, intrapartum pyrexia, and/or PROM. Other indications that could perhaps be added to this group of clinical indications include COVID-19 or FD/MSL. FD can be due to acute chorioamnionitis or to other non-inflammatory processes as such as abruption, cord events, etc. While the mechanistic implications of COVID-19 on adverse outcomes are primarily due to massive perivillous fibrin deposition and chronic histiocytic intervillositis [14], inflammation has also been previously described in COVID-19-affected placentas [15,16]. Notably, no significant difference in diagnostic performance even if COVID-19 and/or FD/MSL are added as indications for a preliminary UC evaluation.

Accordingly, it is highly feasible to submit a preliminary block of only the UC to allow rapid processing of the cord and a microscopic examination within 24 h or less, while awaiting optimal fixation of the placental disc and extraplacental membranes, to see if there is an FIR. Indeed, Katzman et al. reported that cord inflammation is often present at the early stages of intraamniotic infection [13]. However, acute inflammation in the UC can be contiguous, discrete, or multifocal [17] and, as the Amsterdam sampling protocol examines only two UC cord sections, it may not be surprising that about 25% of cases showed acute chorionic vasculitis without a funisitis in the present study. Likewise, we observed about half of the placentas had MIR without a UC FIR. Our study showed that FIR did not occur in the absence of MIR, in keeping with the current view that an FIR is a sequela to MIR, and

rarely occurs in isolation [6]. Interestingly, a recent study showed that cord inflammation could serve as a surrogate marker for histological chorioamnionitis [18].

Histological examination of a preliminary UC section while waiting for the rest of the placenta to be optimally fixed and processed achieves a balance between expediency and clinical utility and can be recommended in busy pathology departments especially in cases with one or more of clinical suspicion of chorioamnionitis, intrapartum pyrexia and/or PROM, to which COVID-19 or FD/MSL could arguably be added. A finding of FIR on the UC would affirm a need to continue antibiotic treatment. For the reasons discussed earlier, a negative UC FIR finding, on the other hand, would not necessarily lead to a discontinuation of the empiric antibiotic therapy although desirable to avoid side effects in the newborn, including contributing to emerging antibiotic resistance, especially in the neonatal intensive care unit settings [19]. In this situation, additional clinical assessment of the newborn, including other routine laboratory biomarkers such as white cell count and C-reactive protein, would be prudent.

**Author Contributions:** Conceptualization, T.Y.K.; methodology, Y.P.W.; investigation and data curation, Y.P.W.; original draft preparation, Y.P.W.; review and editing, T.Y.K.; supervision, T.Y.K.. All authors have read and agreed to the published version of the manuscript.

**Funding:** This research received no external funding.

**Institutional Review Board Statement:** The study was conducted in accordance with the Declaration of Helsinki and approved by the Institutional Review Board of the Women's and Children's Hospital, Adelaide (protocol code 1287A/09/2025 on 4 October 2022).

**Informed Consent Statement:** Patient consent was waived as this was a retrospective study.

**Data Availability Statement:** Not applicable.

**Acknowledgments:** We would like to thank and acknowledge all staffs from the Department of Pathology, SA Pathology, Australia for the help and support.

**Conflicts of Interest:** The authors declare no conflict of interest.

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
