# Peer review of "Changing Laboratory Practice for Early Detection of a Fetal Inflammatory Response: A Contemporary Approach"

_diagnostics, doi:10.3390/diagnostics13030487_

Round 1

Reviewer 1 Report

Dear authors,

You conducted an study regarding the laboratory investigation of fetal inflammatory response. Your methodology is very good. I only have one question.

In line 68 you mention the tests used for investigating differences between variables. Was the distribution of all your variables normal? Which test did you use to examine the normality of distribution of your continuous variables. If you didn' t examine it, can you explain why?

The presentation and interpretation of your results are excellenty presented in your manuscript.

Reviewer 2 Report

Changing Laboratory Practice for Early Detection of a Fetal Inflammatory Response: A Contemporary Approach by Yin Ping Wong and T Yee Khong.

The article is interesting for audience and the conclusion is crucial in prompting the pathological analysis of placenta and UC. This is likely to increase the antibiotics use in newborns: this requires careful monitoring. The references require careful check. The article is suitable for publication.
